# Molecular Insights into the Superiority of Platelet Lysate over FBS for hASC Expansion and Wound Healing

**DOI:** 10.3390/cells14151154

**Published:** 2025-07-25

**Authors:** Sakurako Kunieda, Michika Fukui, Atsuyuki Kuro, Toshihito Mitsui, Huan Li, Zhongxin Sun, Takayuki Ueda, Shigeru Taketani, Koichiro Higasa, Natsuko Kakudo

**Affiliations:** 1Department of Plastic and Reconstructive Surgery, Kansai Medical University, 2-5-1 Shin-machi, Hirakata 573-1010, Osaka, Japan; 2Department of Plastic and Reconstructive Surgery, Kansai Medical University Medical Center, 10-15 Fumizono-cho, Moriguchi 570-8507, Osaka, Japan; 3Department of Genome Analysis, Institute of Biomedical Science, Kansai Medical University, 2-5-1 Shin-machi, Hirakata 573-1010, Osaka, Japan

**Keywords:** human adipose-derived stem cells, platelet lysate, cell proliferation, stress resistance, antiaging effects, RNA sequencing

## Abstract

Human adipose-derived stem cells (hASCs) are widely used in regenerative medicine due to their accessibility and high proliferative capacity. Platelet lysate (PL) has recently emerged as a promising alternative to fetal bovine serum (FBS), offering superior cell expansion potential; however, the molecular basis for its efficacy remains insufficiently elucidated. In this study, we performed RNA sequencing to compare hASCs cultured with PL or FBS, revealing a significant upregulation of genes related to stress response and cell proliferation under PL conditions. These findings were validated by RT–qPCR and supported by functional assays demonstrating enhanced cellular resilience to oxidative and genotoxic stress, reduced doxorubicin-induced senescence, and improved antiapoptotic properties. In a murine wound model, PL-treated wounds showed accelerated healing, characterized by thicker dermis-like tissue formation and increased angiogenesis. Immunohistochemical analysis further revealed elevated expression of chk1, a DNA damage response kinase encoded by CHEK1, which plays a central role in maintaining genomic integrity during stress-induced repair. Collectively, these results highlight PL not only as a viable substitute for FBS in hASC expansion but also as a bioactive supplement that enhances regenerative efficacy by promoting proliferation, stress resistance, and antiaging functions.

## 1. Introduction

Stem cells are a unique population characterized by their self-renewal capacity, long-term viability, and multilineage differentiation potential. Among them, human adipose-derived stem cells (hASCs) have garnered significant attention because of their ability to differentiate into various lineages, including adipose tissue, bone, cartilage, tendons, nerves, and fat, under lineage-specific cultivation conditions [1,2]. Compared with other types of mesenchymal stem cells (MSCs), hASCs are widely utilized in the field of reconstructive and aesthetic surgery because of their ease of harvest and ability to yield a significantly greater number of cells [3,4]. Furthermore, hASCs secrete a diverse range of bioactive mediators, including growth factors, cytokines, and extracellular vesicles such as exosomes and microvesicles, which play crucial roles in regenerative medicine [5]. These soluble mediators, including vascular endothelial growth factor (VEGF), hepatocyte growth factor (HGF), fibroblast growth factor (FGF), insulin-like growth factor-1 (IGF-1), brain-derived neurotrophic factor (BDNF), and interleukins (e.g., IL-1Ra, IL-6, and IL-8), regulate immune responses through paracrine mechanisms [6]. These properties make hASCs particularly effective for applications such as wound healing [7], alopecia treatment [8], skin rejuvenation [6], and fat grafting [9]. For the clinical application of hASCs, many cells with high stemness potential are needed. To expand hASCs efficiently, specific culture supplements are commonly used, with fetal bovine serum (FBS) being the most representative. However, the risks of xenoimmunization against bovine antigens and transmission of pathogens have necessitated the development of human-derived alternatives for cell therapy protocols [10]. Therefore, platelet-derived products, particularly platelet lysate (PL) and platelet-rich-plasma (PRP), have gained attention as substitutes for FBS [11].

PL is a platelet-derived product prepared through repeated freeze‒thaw cycles and sonication from fresh blood or outdated platelet concentrates. In recent years, the clinical use of PL has increased because of its acellular nature, which reduces concerns about immunogenicity, and its high concentrations of growth factors and cytokines, including platelet-derived growth factor (PDGF), FGF, epidermal growth factor (EGF), and transforming growth factor-β (TGF-β) [10,12]. For cell expansion, commercially available PL is often used, as it is manufactured under strict Good Manufacturing Practices (GMPs), primarily from blood bank sources. GMP-grade PL eliminates the possibility of the risk of allogenic blood-derived infection and leads to reproducible clinical and research outcomes [12]. Clinically, PL therapy has demonstrated positive outcomes in the treatment of osteoarthritis, bone regeneration, ocular disease, infertility, and wound healing [7,13,14,15].

Numerous in vitro studies have shown that culturing human dermal fibroblasts, human umbilical vein endothelial cells (HUVECs), and MSCs with PL or PRP results in a significantly greater growth rate, cellular viability, and differentiation potential than does culturing with FBS [16,17,18,19]. We previously reported that PL or PRP promoted cell adhesion, migration, and proliferation via multiple signaling pathways, including the AKT and ERK1/2 pathways [20,21]. However, the advantages of PL for hASC expansion have not been fully elucidated. In this study, we conducted a comprehensive analysis of PL-cultured hASCs via RNA sequencing (RNA-seq). This analysis confirmed the proliferative effects of PL on hASCs and revealed its role in enhancing cellular stress resistance and antiaging properties.

## 2. Materials and Methods

### 2.1. Isolation and Culture of hASCs

Human adipose tissue was harvested from the abdominal subcutaneous tissues of patients who had undergone plastic surgery. hASCs were isolated as previously described [16,22]. The obtained cells were cultured with Dulbecco’s modified Eagle’s medium (DMEM; FUJIFILM Co., Tokyo, Japan), 10% fetal bovine serum (FBS; HyClone^TM^, Cytiva, Marlborough, MA, USA), and penicillin/streptomycin (complete medium). The cells reached 80–90% confluence after 6–7 days. The hASCs were subsequently stored in a liquid nitrogen tank. Cells from passages 6–9 were used for all experiments. The expression of the cell surface antigens CD73 and CD90 in the hASCs used in this study has been previously reported [21]. This study was approved by the Ethics Review of Kansai Medical University in accordance with the ethical guidelines of the Helsinki Declaration of 1975 (approval code: 2023174). All participants enrolled in this study provided informed consent.

### 2.2. Cell Proliferation Assay

hASCs were seeded into 96-well cell culture plates at a density of 3 × 10^3^ cells per well and incubated at 37 °C. The cell culture medium was then replaced with serum-free DMEM (control medium), DMEM supplemented with 10% FBS, and DMEM supplemented with 3% or 5% PL. In all experimental sections of this study, PL refers to the commercial pooled human platelet lysate PLTMax^®^ (Merck Millipore, Darmstadt, Germany). After incubation for 48 h, cell proliferation was examined via a Cell Counting Kit-8 (CCK-8; Dojindo Molecular Technology, Kumamoto, Japan) according to the manufacturer’s instructions, as described previously [16,20,22]. The optical absorbance was measured at 450 nm via a multiwell plate reader (EnSpire 2300 Multilabel Reader; PerkinElmer, Waltham, MA, USA).

### 2.3. RNA Sequencing

Total RNA was isolated from hASCs cultured with the control medium, DMEM supplemented with 10% FBS, or DMEM supplemented with 3% PL at 37 °C for 48 h via Maxwell RSC SimplyRNA Cells (Promega Co., Madison, WI, USA). Library construction and sequencing were performed by Macrogen (Tokyo, Japan). The raw reads were trimmed with Trimmomatic v0.35 using the LEADING: 30, TRAILING: 30, SLIDINGWINDOW: 4: 15, and MINLEN: 60 options to maintain high-quality sequences, and the transcript abundances were quantified via kallisto v0.46.0, with 100 bootstrap replicates using the reference transcripts in the human genome GRCh38 (Ensemble release 93). Differentially expressed transcripts were identified via the Wald test of the Sleuth R package v0.30.0, with a q-value < 0.05 after multiple testing correction (Benjamini–Hochberg method). The q-value downstream multivariate analyses were performed in R (v4.3.3); principal-component analysis (PCA) and hierarchical clustering (Euclidean distance and Ward’s linkage) of log2-transformed TPM values were used to generate the PCA plot and dendrogram. The RNA-seq data have been deposited with links to the BioProject accession number PRJDB20462 in the DDBJ BioProject database.

### 2.4. Quantitative Reverse Transcription‒Polymerase Chain Reaction (RT‒qPCR)

Total RNA was isolated from hASCs cultured under conditions similar to those used for RNA sequencing via a Maxwell RSC kit (Promega Co.). RT‒qPCR was performed via the SYBR Green RT‒qPCR Master Mix (Qiagen GmbH, Venlo, The Netherlands) according to the manufacturer’s protocol on a Rotor-Gene Q3 Real‒Time PCR System (Qiagen GmbH). The PCR thermocycling conditions were 40 cycles of 10 s at 95 °C and 20 s at 60 °C. Relative gene expression changes were calculated via the 2^‒ΔΔCq^ method with glyceraldehyde 3-phosphate dehydrogenase (GAPDH) as the internal reference gene. The PCR primers used in the present study are listed in Appendix A.

### 2.5. Stress Resistance Assay

hASCs were seeded into a 96-well cell culture plate at a density of 3.0 × 10^3^ cells/well and incubated in DMEM with 10% FBS or 3% PL at 37 °C for 48 h. Then, to evaluate DNA damage and oxidative stress, the cell culture medium was replaced with serum-free DMEM (control medium), and the cells were treated with 5-fluorouracil (5-FU; FUJIFILM Co.; for DNA damage) and hydrogen peroxide (H_2_O_2_; for oxidative stress) for 2 h. After exposure, the cells were further incubated in DMEM supplemented with 10% FBS or 3% PL for 24 h at 37 °C. The cell viability was then examined via a CCK-8 assay.

### 2.6. Immunofluorescence Assay

The cells were incubated in DMEM supplemented with 10% FBS and 3% PL. The culture medium was then replaced with control medium. The cells were then treated without or with 100 µg/mL 5-FU or 10 µM H_2_O_2_ for 1 h, followed by incubation with DMEM supplemented with 10% FBS and 3% PL. After the cells were washed with PBS, they were fixed with 4% paraformaldehyde for 20 min and then permeabilized in 0.3% Triton X-100 in PBS for 20 min. The cells were incubated with antibodies against γH2AX (S-139) (Cusabio Technology LLC., Houston, TX, USA), followed by incubation with FITC-conjugated goat anti-rabbit IgG (1:100) (Proteintech, Rosemont, IL, USA) [21,22]. Images of the antigens were captured via a Keyence BZ-9000 fluorescence microscope (Keyence, Osaka, Japan).

### 2.7. Apoptosis Assay Using Annexin V

hASCs were incubated in DMEM supplemented with 10% FBS or 3% PL, and the culture medium was replaced with control medium. The cells were then exposed to 100 µg/mL 5-FU and 10 µM H_2_O_2_. The cells were incubated with DMEM supplemented with 10% FBS or 3% PL at 37 °C for 24 h. After trypsinization, each cell suspension (100 µL) was transferred to a 1.5 mL microtube. Then, 100 µL of Muse Annexin V & Dead Cell solution (Merck Millipore, Darmstadt, Germany) was added to each tube, mixed, and incubated for 20 min at room temperature. The percentages of the cell populations were analyzed via a Muse Cell Analyzer (Merck Millipore, Burlington, MA, USA).

### 2.8. Senescence β-Galactosidase (SA-β-gal) Assay

hASCs were seeded into 96-well cell culture plates at a density of 5.0 × 10^3^ cells per well and incubated with DMEM containing 10% FBS or 3% PL at 37 °C. The cells were treated with 0.2 µM doxorubicin, an aging-inducing drug, for 24 h. The density of the target cells was examined via crystal violet staining [23]. After cell lysis, SA-β-gal activity was measured fluorometrically according to the protocol of an S-gal kit (Dojindo Molecular Technology). The fluorescence intensity was examined via an EnSpire 2300 Multilabel Reader (PerkinElmer). Separately, the doxorubicin-treated cells were washed with PBS and fixed with 2% paraformaldehyde. The cells were incubated with the SA-β-gal substrate at pH 6.0 [23]. The fluorescence images of SA-β-gal in cells were also captured by a Keyence BZ-9000 fluorescence microscope (Keyence).

### 2.9. Murine Wound Repair

C57bl6J/Jcl male mice (n = 21 total 8–9-week-old mice) were anesthetized via isoflurane (Wako Pure Chemical Industries Ltd., Osaka, Japan). Before surgical procedures, the hair on the backs of the mice was shaved with an electric razor (Thrive; Daito Electric Machine Ind. Co., Ltd., Osaka, Japan) and depilated with depilating cream (Kracie, Tokyo, Japan). Using a punch biopsy tool (Kai Industries Co., Ltd. Gifu, Japan) and Iris scissors, a full-thickness wound 6 mm in diameter, including the panniculus carnosus, was made. Collagen/Gelatin Sponges (CGSs) (2 × 2 cm; Pelnac Gplus^®^, Gunze Co. Ltd., Ayabe, Japan) treated with saline (for the control group) or PL were dressed onto each wound. To ensure the effectiveness of these treatments, a donut-shaped silicone skin splint (Fuji System Corp, Tokyo, Japan) was placed onto the CGS and stitched with black nylon 5-0 (Bear Corporation, Osaka, Japan) [24]. Digital photographs were taken at 0, 3, and 7 days to evaluate the differences in the wound healing process between the groups. The wound area was evaluated by tracing the margin of the digital photograph with the ImageJ 1.53i software program (National Institutes of Health, Bethesda, MD, USA). The wound areas on days 3 and 7 were then compared with the original area of the wound on day 0, with the results presented as a percentage of the remaining wound area (%).

The animal experiments were performed in accordance with protocols approved by the Animal Care and Use Committee of Kansai Medical University. The work has been reported in line with the ARRIVE guidelines 2.0, and the animal protocols were based on the “Guide for the Care and Use of Laboratory Animals, Eighth Edition,” published by the National Research Council of the National Academies in 2011. All animal experiments were approved by the Animal Care and Use Committee of Kansai Medical University. The approved project title was “Characterization and clinical significance of adipose-multi-potential cells.” The approval number is 24-123 (1), and the approval date was 22 May 2024.

### 2.10. Histological and Immunochemical Analysis of Wound Healing

The mice were sacrificed via carbon dioxide inhalation on days 3 and 7 after surgery (n = 3 in each group). The wounds were dissected and fixed in 10% formalin-buffered solution (Wako Pure Chemical Industries, Ltd.) for histological examination. Paraffin sections were prepared and stained with hematoxylin and eosin (HE) [24]. The sections were separately rinsed in PBS and incubated with anti-chk1 (Cusabio Technology) at a 500-fold dilution at 4 °C overnight. The sections were then rinsed in PBS, followed by incubation with HRP-labeled anti-rabbit immunoglobulin (Dako Japan Co., Kyoto, Japan) at room temperature for 30 min. The sections were rinsed in PBS and exposed to diaminobenzidine. After counterstaining with hematoxylin, histologic photographs were taken and analyzed via a Nanozoomer 2.0 HT whole-slide scanner with the NDP.view2 software program (Hamamatsu Photonics, Hamamatsu, Japan) at 40× magnification.

### 2.11. Statistical Analysis

The data are expressed as the means ± standard deviations (SDs) (n = 3/4). All the statistical analyses were performed via JMP^®^ Pro v.17.2.0 (JMP Statistical Discovery, LLC, SAS Campus Drive Cary, NC, USA). Data homogeneity was examined via the Shapiro–Wilk test. For comparisons between two groups, Student’s *t* test was used. Significant differences among multiple groups were evaluated via one-way ANOVA followed by Tukey’s post-hoc test. *p* < 0.05 was considered to indicate a statistically significant difference.

## 3. Results

### 3.1. Proliferative and Stress Resistance Effects of PL on hASCs Cultured with PL

We previously reported that PL enhances the proliferation of hASCs [25]. To confirm the proliferative ability of PL on hASCs, we performed a cell proliferation assay. Compared with that in the control group, cell proliferation markedly increased in both the PL and FBS groups (Appendix A). No significant difference in cell proliferation was observed between the 3% PL and 5% PL groups. Furthermore, the 3% and 5% PL groups exhibited greater cell proliferation than the FBS group. To explore the molecular mechanism underlying the growth advantage of PL, we performed RNA sequencing on hASCs cultured under three conditions: control (serum free), 10% FBS, and 3% PL. First, to obtain an overview of the RNA-seq data from the three culture conditions (control, FBS, and PL), we performed principal-component analysis (PCA; PC1 26.7%, PC2 14.8%) and hierarchical clustering. As shown in Figure 1a,b, samples within each group formed tight clusters, whereas the three groups were clearly separated. These findings indicate high reproducibility among replicates and marked transcriptional differences between treatment conditions. We then carried out differential gene expression analysis. A total of 4336 mRNAs were upregulated in the PL group compared with the control group, whereas 3832 mRNAs were upregulated in the FBS group compared with the control group (q < 0.05) (Figure 1c). A total of 1485 mRNAs were commonly upregulated in both the PL and FBS groups. On the other hand, 4021 mRNAs in the PL group were downregulated compared with those in the control group, whereas 2548 mRNAs in the FBS group were downregulated compared with those in the control group (q < 0.05). A total of 2239 mRNAs were downregulated in both the PL and FBS groups. The results of the clustering analysis of mRNA expression among the three groups (Figure 1d) revealed that the genes whose expression was upregulated in the PL group were the majority related to cell proliferation (q < 1.0 × 10^−7^, β > 4.00), and the genes whose expression was downregulated were involved in the inhibition of cell growth (q < 1.0 × 10^−7^, β > 4.00). The results support our previous studies [25] and the present data (Appendix A). Furthermore, we focused on investigating unique characteristics of gene expression specific to the PL group. We then performed over-representation analysis (ORA) using Gene Ontology (GO) to define the biological process in the PL and FBS groups compared with the control group (*p* < 0.05, b > 4.00) (Figure 1e). As expected, both the PL and FBS groups presented the activation of genes involved in the cell cycle, mitotic cell cycle, cell division, and chromosome segregation. In the PL group, genes involved in stress resistance, such as the cellular response to stress, DNA repair, and the DNA stress response were specifically upregulated. We identified representative genes involved in cell proliferation (CDC20, CEP55, FEN1, MT2A, and TPX2) and stress resistance (BARD1, CHEK1, and RAD18) and detected the highest expression in the PL group (Appendix A). To confirm gene expression, RT‒qPCR analysis was performed. The expressions of genes involved in cell proliferation, including CDC20, FEN1, MCM5, and MT2A, and the stress response genes CHEK1, FANCA, and RAD18, were markedly upregulated in the PL group (Figure 1f). In addition, ORA using KEGG and the Reactome pathway showed significant enrichment of cell cycle/proliferation pathways, particularly in the PL group (Appendix A). These results indicated that the genes involved in the stress response and cell proliferation in PL-cultured hASCs were more potent than those in FBS-cultured cells.

### 3.2. Effect of PL on the Survival of hASCs Treated with Stress-Inducing Agents

On the basis of the RNA-seq data, we examined whether PL protects hASCs from cellular stress. The cells were then treated with 5-FU and H_2_O_2_, which are reagents that disrupt DNA synthesis and induce oxidative stress, respectively. Compared with those in the FBS group, the number of surviving cells in the PL group increased upon exposure to 5-FU (5, 20, and 100 µg/mL) (Figure 2a). When the cells were treated with H_2_O_2_, the resistance of the PL group was greater than that of the FBS group at all concentrations (Figure 2b). Thus, hASCs cultured with PL were resistant to cellular and DNA damage.

### 3.3. Effect of PL on the Apoptosis of hASCs Treated with Stress-Inducing Agents

An increase in the phosphorylation of histone H2A. X is known to be associated with DNA damage. To examine the effect of DNA damage under stress conditions, the phosphorylation of histone H2A. X was examined. The level of H2A. X phosphorylation in the PL-treated group was lower than that in the FBS group (Figure 3a,b). We next examined the effect of PL on the apoptosis of hASCs upon exposure to H_2_O_2_ or 5-FU. When the cells were treated with 5-FU, total apoptosis was 15.5% in the FBS group and 6.6% in the PL group. Similarly, after treatment with H_2_O_2_, total apoptosis was 14.6% in the FBS group and 6.6% in the PL group (Figure 4a,b). Thus, PL contributes to protection against apoptosis induced by DNA damage.

### 3.4. Effect of PL on Doxorubicin-Induced hASC Aging

RNA-seq analysis revealed that the mRNA levels of p21 and MMP2, which are senescent markers, in the PL group were lower than those in the FBS group (Appendix A). We next examined the antiaging effect of PL. hASCs were treated with doxorubicin, a drug that induces cellular senescence, and the activity of senescence-associated β-galactosidase (SA-β-gal), a marker of cellular senescence, was examined. In the fluorescence images, the cells treated with PL presented lower SA-β-gal staining than those treated with FBS (Figure 5a). The specific β-gal activity per cell density in PL-cultured cells was lower than that in FBS-cultured cells (Figure 5b). In addition, when the expressions of the inflammatory markers IL-1β and IL-6 in hASCs were examined, the levels of both mRNAs associated with PL were lower than those associated with FBS (Appendix A). These results indicated that PL suppressed the aging of hASCs.

### 3.5. Effect of PL on Wound Healing In Vivo

We finally examined the effect of PL on mouse wound healing in vivo. Figure 6a,b show representative wound images of the two groups on days 0, 3, and 7 and the average wound area. On day 3, no significant difference in the wound area was observed between the two groups (control: 37.7 cm^2^, PL: 31.1 cm^2^), but on day 7, the wound area in the PL group was markedly smaller than that in the control group (control: 27.6 cm^2^, PL: 8.3 cm^2^). Thus, PL promoted faster wound contraction than did the control.

### 3.6. Histological and Immunohistochemical Analysis of the Wound Healing Effect of PL

The histological morphology of the wound tissue on days 0, 3, and 7 is shown in Figure 7a. When the thickness of the newly formed dermis-like tissue was measured on day 3, the relative thickness in the PL group was significantly greater than that in the control group. On day 7, wound contraction occurred, and well-formed granulation tissues and numerous blood vessels were found in the deeper layers of the PL group. These results indicated that, compared with the control, PL promoted the formation of thicker dermis-like tissue during the early stages of wound healing and angiogenesis in deeper layers. Immunohistochemical staining for chk1, a marker of cell division and the stress response, was performed to investigate the cellular response to stress (Figure 7b). The cells with positive antibody reactions exhibited intense nuclear staining. On day 3, the PL group presented thicker dermis-like tissue in which a greater number of chk1-positive cells were found. On day 7, the number of chk1-positive cells within the dermis-like tissue in the PL group was significantly greater than that in the FBS group. Furthermore, positively stained cells in the PL group exhibited markedly intense nuclear staining. These results indicate that PL induces a stress-responsive effect on cells and promotes cell division, thereby contributing to improved wound healing.

## 4. Discussion

The present study demonstrated that, compared with FBS, PL significantly enhances the proliferation and stress resistance of hASCs. Our findings indicate that hASCs cultured with PL exhibited superior cell growth, stress resistance, and antiaging properties, highlighting the potential clinical advantages of PL as a culture supplement. The ability of PL to enhance superproliferation and defend against cellular stress indicates the potential for hASC expansion. Thus, we provide valuable insights into the molecular mechanisms underlying the regenerative effects of PL and its possible applications in cell therapy and regenerative medicine.

FBS is the most common serum additive for in vitro use, supporting cell adhesion, growth, and proliferation across various cell types. On the basis of the RNA-seq data, the expression of genes involved in the cell cycle, mitotic cell cycle, cell division, and chromosome segregation increased in the FBS-supplemented hASCs. Thus, while FBS is an effective supplement for culturing hASCs, it carries risks, such as the transmission of known and unknown pathogens and immunological reactions. The RNA-seq analysis also revealed that genes associated with cell division were significantly upregulated and widely distributed in cultures with PL. These findings align with previous observations that PRP or PL enhances the proliferation of hASCs compared with FBS [16,25,26]. The combination of growth factors abundant in PLs, including PDGF, VEGF, and TGF-β, markedly enhances the proliferation of hASCs [22]. We confirmed the superiority of PL at the molecular level through the upregulation of various genes associated with cell cycle progression and proliferation. The changes in gene expression involved in cell growth were consistent with findings from previous in vivo studies on the proliferative effects of PL [15,27]. In addition to the upregulation of proliferative genes, the expression of genes related to stress resistance, such as those involved in cellular response to stress, DNA repair, and DNA stress response, particularly increased in the PL group. This finding was confirmed by RT‒qPCR analysis, which revealed that stress resistance genes, such as CHEK1, RAD18, and BARD1, were highly expressed in PL-treated hASCs (Appendix A). These results suggest that PL not only promotes cell division but also enhances cellular defense mechanisms against oxidative and genotoxic stress. PLs provide a more supportive environment for cell survival and proliferation while reducing concerns related to heterologous immune responses and pathogen transmission.

Understanding the molecular basis of wound healing enables the identification of endogenous recovery pathways naturally activated following injury, facilitating the development of therapeutic strategies to enhance recovery in cases of impaired wound healing. Widyaningrum et al. reported that PL protected corneal endothelial cells against tert-butyl hydroperoxide-induced oxidative stress [28]. They reported that the cells cultured with PL presented increased expression of glutathione S-transferase, an antioxidant protein, as well as the antiapoptotic proteins BCL-2 and BCL-XL. PL exerts a protective effect against hypoxia-induced apoptosis in SV-HUC-1 cells by regulating oxidative stress and the mitochondria-mediated intrinsic apoptotic pathway [29]. PRP, or platelet-rich fibrin lysate, another platelet-derived product, protects against ultraviolet-A-induced damage to human dermal fibroblasts and melanocyte-derived cells [30]. In contrast, PL promoted osteoblast proliferation but did not promote human dental pulp cell proliferation or inhibit inflammation under stress conditions [31]. Furthermore, PRP induces an inflammatory response in tendon fibroblasts, leading to the generation of reactive oxygen species and the activation of oxidative stress pathways, and it does not appear to significantly influence macrophage polarization [32]. The observed differences between the effects of PL and PRP may be attributed to variations in the cell types used and differences in platelet preparation methods. Nevertheless, findings from in vitro studies on the effects of platelet-derived nutrients remain inconclusive. To address this gap, we conducted a systematic comparative analysis of gene expression profiles and cellular functions following PL supplementation. Compared with those cultured with FBS, hASCs cultured with PL exhibited significantly greater survival rates following exposure to 5-FU and H_2_O_2_ (Figure 2a,b). This increased resistance to cellular damage was supported by immunofluorescent analysis, which revealed reduced phosphorylation of histone H2A. X, a marker of DNA damage, in PL-treated cells (Figure 3a,b). Additionally, apoptosis assays revealed that PL suppressed the apoptosis induced by DNA damage, suggesting its potential role in enhancing cell survival under adverse conditions (Figure 4a,b). Therefore, gene expression specialized for cellular stress was further confirmed by the protective effects of PL on cell survival under chemically induced stress conditions. Our experimental findings further support these previous studies, reinforcing the potential application of PL in cell-based therapies where cell resilience is a critical factor.

Senescence, a hallmark of aging, is a cellular state characterized by irreversible cell cycle arrest and the secretion of proinflammatory molecules into the surrounding microenvironment, a phenomenon known as the senescence-associated secretory phenotype [33]. As an organism ages, senescent cells accumulate, leading to decreased cellular proliferation and compromised tissue regeneration and function [34]. Oxidative stress and DNA damage contribute to cellular senescence, which is widely recognized by the accumulation of markers such as p16, p21, and SA-β-gal [35,36]. The RNA-seq data revealed a decrease in p21 and MMP2 gene expression in the PL-cultured group compared with the FBS group (Appendix A), prompting further investigation into cellular senescence. We then found that the expression of SA-β-gal in PL-cultured hASCs under chemically aging-induced conditions was significantly lower than that in FBS-cultured cells (Figure 5a,b). These results suggest that PL not only supports cell proliferation but also contributes to maintaining cellular youthfulness by reducing senescence-associated inflammation. Other investigators reported that human articular chondrocytes cultured with PL exhibited delayed senescence via the BMP-TAK1-p38 pathway, and that PRP promoted the transcription of SIRT1, a longevity gene, in fibroblasts, hence delaying their senescence [37,38]. Taken together, these findings indicate that PL is a promising candidate for applications in regenerative medicine, particularly in aging-related disorders.

PL has been used in clinical trials, and the therapeutic efficacy of platelet-derived substitutes has been reported in various medical conditions, including diabetic ulcers, alopecia, and osteoarthritis [39,40,41], thus promising results in treating chronic wounds. Experimental studies in animal models have demonstrated that the local application of platelet-derived substitutes, including PL, induces granulation tissue formation and re-epithelialization and thus accelerates wound repair through the interaction of various growth factors [25,42,43]. On the basis of these findings, we investigated the wound healing effects of PL and further explored its underlying mechanisms on the basis of the results of RNA-seq analysis, with a particular focus on CHEK1. CHEK1 encodes checkpoint kinase 1 (chk1), a critical component of the DNA damage response in human cells [44]. Chk1 maintains genomic stability by ensuring proper chromosome alignment, preventing chromosome mis-segregation, and regulating the completion of cytokinesis, thereby facilitating accurate cell division [45]. Compared with the control group, the PL-treated group exhibited faster wound closure and greater dermal thickness (Figure 6a,b). Histological analysis further revealed enhanced granulation tissue formation and angiogenesis, as evidenced by increased chk1-positive cells in PL-treated wounds (Figure 7a,b). These findings suggest that the enhanced wound healing observed in the PL-treated group may be accompanied by parallel activation of proliferative and stress defense regulators, at least in part through the activation of chk1-dependent mechanisms.

Several limitations should be noted. First, all platelet lysate preparations were restricted to the pooled, GMP-grade product PLTMax^®^, and the efficacy of autologous or laboratory-prepared PL remains to be confirmed. Second, PLTMax^®^ was applied xenogeneically in the murine model; validation using mouse-derived PL—or other syngeneic systems—will be required to strengthen translatability. Third, the in vivo experiments were performed exclusively in male mice, and potential sex-dependent differences should be assessed in future studies that include female animals.

Although this study revealed the adjuvant role and mechanism of PL in wound healing, further investigations are needed to determine the key factors driving these pathways and the synergistic relationship between PL and hASC proliferation and differentiation in wound repair.

## 5. Conclusions

Our findings provide strong evidence supporting the use of PL as an alternative to FBS for hASC expansion and regenerative applications. PL not only enhances proliferation but also confers stress resistance and antiaging effects, making it a highly beneficial culture supplement for stem cell-based therapies. Further investigations are needed to elucidate the precise molecular pathways involved and to optimize PL formulations for clinical use. These insights may contribute to the development of safer and more effective regenerative medicine strategies.

## Figures and Tables

**Figure 1 cells-14-01154-f001:**
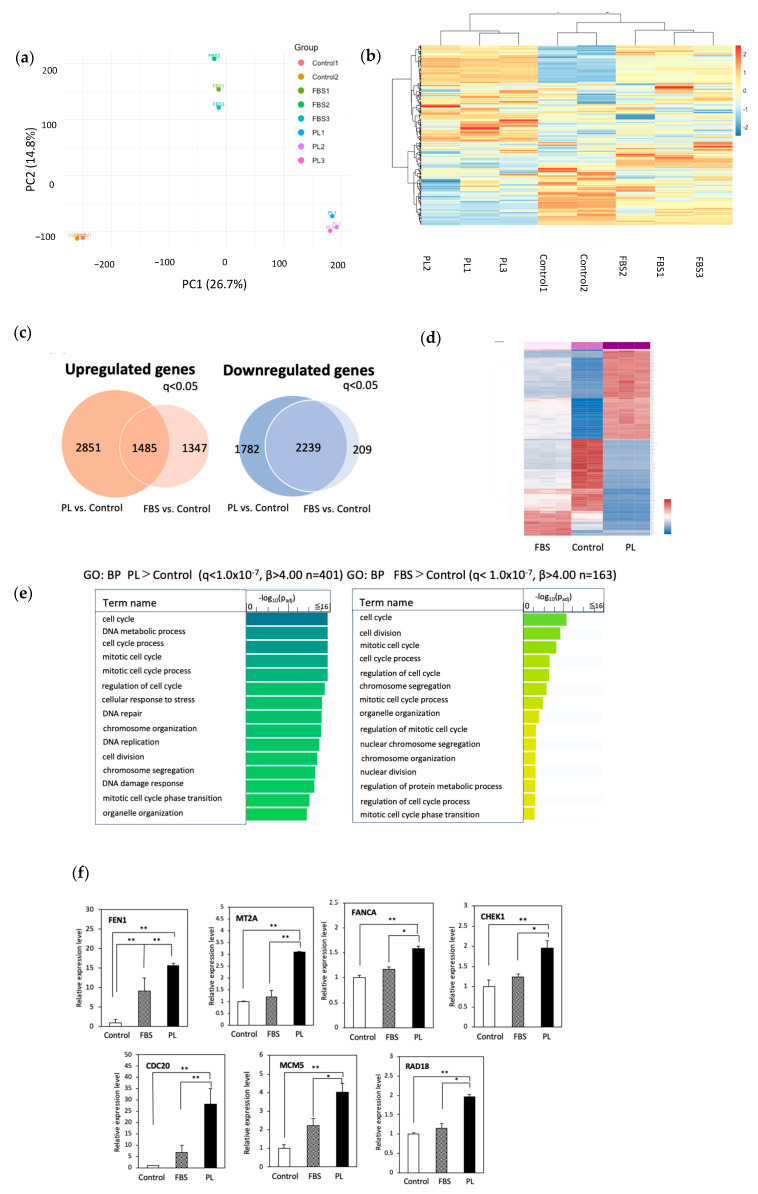
Analysis of RNA sequencing data from hASCs cultured without or with 10% FBS or 3% PL. hASCs were cultured with control (serum free), 10% FBS, and 3% PL-containing DMEM for 48 h. RNA was extracted and used for sequencing. (**a**) Principal-component analysis (PCA). PCA was performed on log2-transformed TPM values for the 500 most variable genes. Axes denote PC1 (26.7 %) and PC2 (14.8 %). Each point represents one biological replicate (control, n = 2; and FBS and PL, n = 3 per group). (**b**) Hierarchical clustering heat-map of the 500 most variable genes in the control, FBS, and PL groups. Rows are genes and columns are individual samples; red and blue indicate relative up- and downregulation (row-wise z-scores). Dendrograms were generated with Euclidean distance and Ward’s linkage. (**c**) Venn diagram of the number of differentially expressed genes (q < 0.05) in the PL and FBS groups compared with the control group. (**d**) Pearson correlation (heatmap) of the mRNA levels in the control, FBS, and PL groups. The red and blue colors on the y axis represent up- and downregulated genes, respectively. (**e**) Gene Ontology analysis of upregulated genes (q < 1.0 × 10^−7^, β > 4.00). P_adj_ indicates the *p*-values corrected for multiple testing via the Benjamini–Hochberg method. (**f**) RT‒qPCR of upregulated mRNAs in hASCs. Relative expression levels of genes related to cell proliferation (FEN1, MT2A, CDC20, and MCM5) and stress resistance (CHEK1, RAD18, and FANCA). The values are expressed as the means ± SDs (n = 3) * *p* < 0.05, ** *p* < 0.01 between the indicated groups.

**Figure 2 cells-14-01154-f002:**
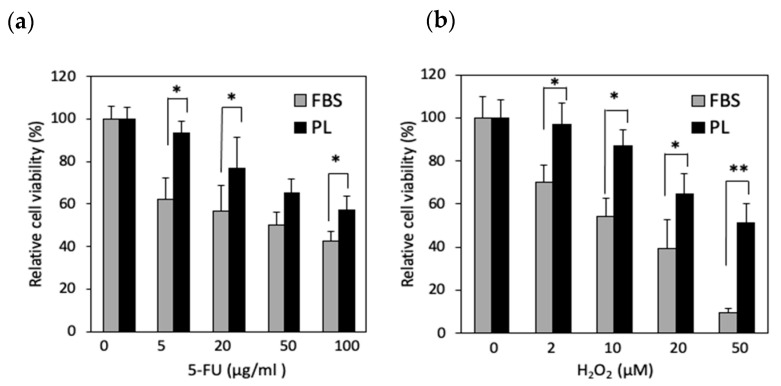
Effect of PL on surviving hASCs treated with 5-FU or H_2_O_2_. The cells were cultured with FBS- and PL-containing media and then exposed to 5-FU (**a**) and H_2_O_2_ (**b**) for 1 h. The values are expressed as the means ± SDs (n = 4) * *p* < 0.05, ** *p* < 0.01 between the indicated groups.

**Figure 3 cells-14-01154-f003:**
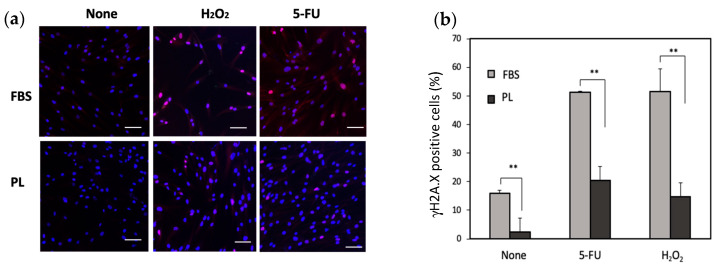
Phosphorylation of histone H2A. X (γH2A. X) in hASCs treated with 100 µg/mL 5-FU or 10 µM H_2_O_2_. (**a**) Images of the γH2A. X-positive cells were captured via a fluorescence microscope. Bars = 100 µm. (**b**) Quantitative analysis of γH2A. X-positive cells. A total of 300 cells were counted under a microscope. The data are expressed as percentages of γH2A. X-positive total cells (n = 3); ** *p* < 0.01 versus the control.

**Figure 4 cells-14-01154-f004:**
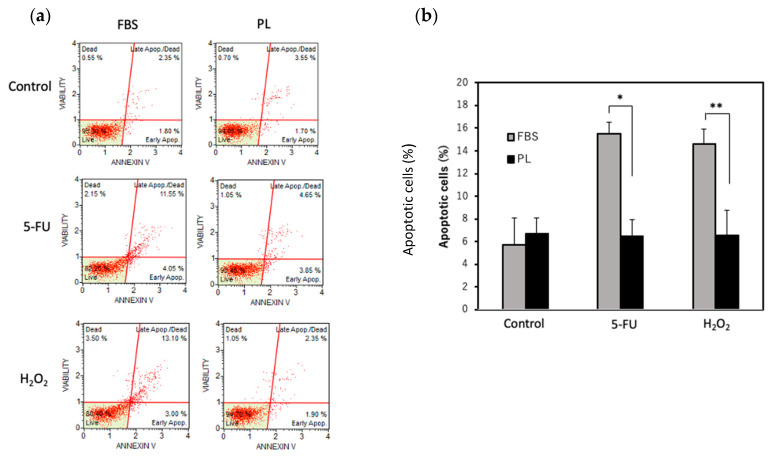
Effect of PL on the apoptosis of hASCs treated with 5-FU or H_2_O_2_. Total apoptosis in hASCs treated with 100 µg/mL 5-FU or 10 µM H_2_O_2_ was evaluated via an annexin V assay. (**a**) Scatchard plots of the results of the apoptosis assay. (**b**) Quantitative analysis of total apoptotic cells after treatment with 5-FU or H_2_O_2_. The values are expressed as the means ± SDs (n = 3). * *p* < 0.05 and ** *p* < 0.01 versus the control.

**Figure 5 cells-14-01154-f005:**
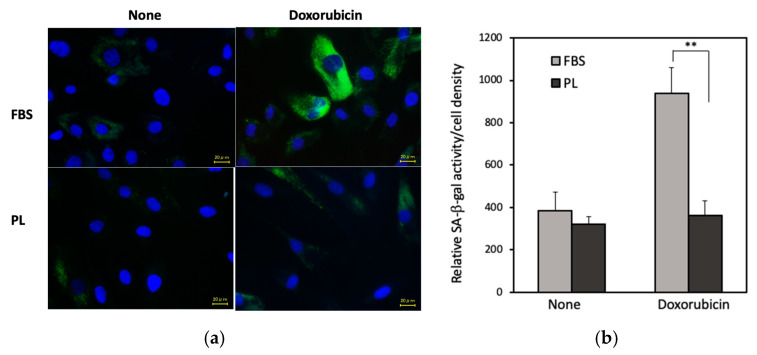
Effect of PL on the doxorubicin-induced aging of hASCs. (**a**) Fluorescence images of SA-β-gal in hASCs. The cells cultured with 3% PL or 10% FBS were treated with 0.2 µM doxorubicin for 24 h. After the cells were fixed with paraformaldehyde, SA-β-gal was examined and observed via fluoromicroscopy. Bars = 20 μm. (**b**) Quantitative analysis of the level of SA-β-gal activity normalized to the hASC density. The values are expressed as the means ± SDs of four experiments. ** *p* < 0.01 versus the control.

**Figure 6 cells-14-01154-f006:**
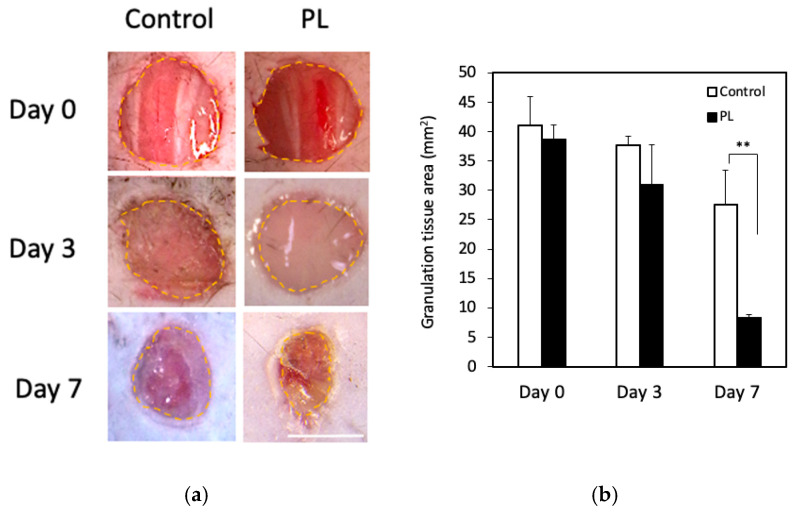
Effect of PL on wound healing in vivo. (**a**) Macroscopic images of control and PL-treated wound areas on days 0, 3, and 7. The yellow broken line indicates the wound margin. Bar = 2.0 mm. (**b**) Comparison of the average wound area. The wound areas on days 3 and 7 were compared with the original area of the wound on day 0. Data are presented as a percentage of the remaining wound area (each group: n = 5). ** *p* < 0.01 versus the control.

**Figure 7 cells-14-01154-f007:**
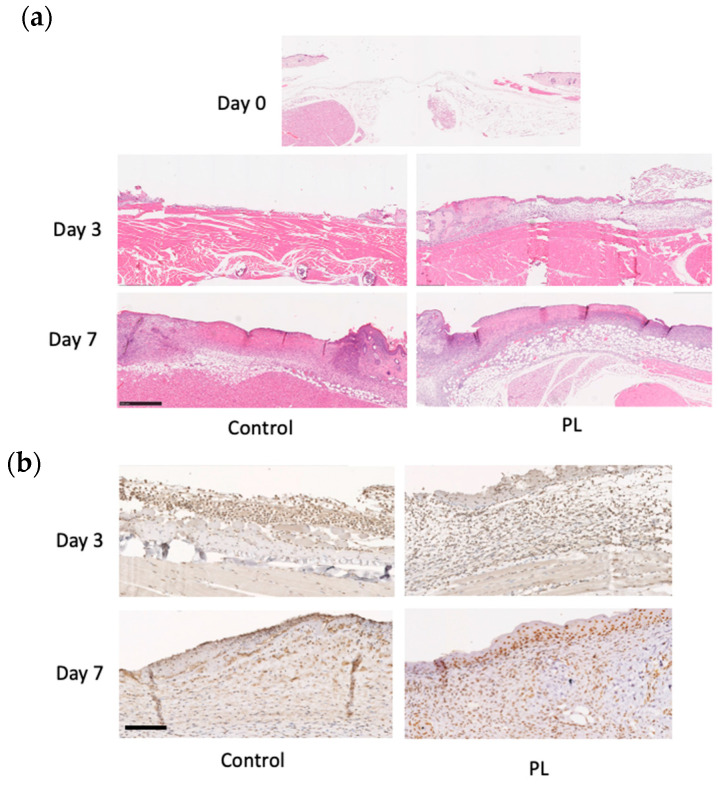
Histological and immunohistochemical analysis of mouse wound healing. (**a**) Representative HE staining of skin samples from mice treated with or without PL at days 0, 3, and 7. The arrows indicate the position of the capillaries. Bar = 500 µm. (**b**) Representative immunohistochemical chk1 staining of skin wound samples from mice treated without or with PL at days 3 and 7. Bars = 500 µm.

## Data Availability

The raw RNA-seq data of MSCs have been deposited with links to the BioProject accession number PRJDB20462 in the DDBJ BioProject database. No additional datasets were generated or analyzed during the current study.

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
