# Peer review of "Molecular Insights into the Superiority of Platelet Lysate over FBS for hASC Expansion and Wound Healing"

_cells, 2025, doi:10.3390/cells14151154_

Round 1

Reviewer 1 Report

Comments and Suggestions for Authors

Suggestion:minor revisions

  1. Lines 71-74, your references are relevant but old. There is plenty evidence in the last 2-3 years that confirm these lines. Please add 2 more recent references in addition to your existing one – to showcase your current knowledge on the subject.
  2. Section 2.5 – do you mean oxidative stress? Use of H2O2 usually refers to oxidative stress test and not called cell damage assay. Please double check this.
  3. Figures 6 and 7 – have you got data to compare control and PL with FBS – as shown in other figures to demonstrate consistency in your research?
  4. Please provide a schematic of the potential mechanism you think PL uses to have the advantages it does over FBS.
  5. Please add a paragraph of limitations to your conclusion – lack of female mice investigation, which implies any findings may be applicable only to cells from male donors; if this approach was to be taken forward.

Author Response

  1. Lines 71-74, your references are relevant but old. There is plenty evidence in the last 2-3 years that confirm these lines. Please add 2 more recent references in addition to your existing one – to showcase your current knowledge on the subject. 

Response1: Thank you for this constructive suggestion. We have incorporated two up-to-date publications that corroborate the points made in lines 71–74, thereby demonstrating our awareness of the most recent literature:

  1. Peters K. et al. Standardized human platelet lysates as adequate substitute to fetal calf serum in endothelial cell culture for tissue engineering. Biomed. Res. Int. 2022, 2022, 3807314.
  2. Wakamoto S. et al. Human platelet lysate produced from leukoreduction filter contents enables sufficient MSC growth. Stem Cell Res. Ther. 2025, 16, 205.

  1. Section 2.5 – do you mean oxidative stress? Use of H2O2usually refers to oxidative stress test and not called cell damage assay. Please double check this.

 Response2: We agree that the original section title was not sufficiently specific. To reflect this, we have renamed Section 2.5. to “Stress Resistance Assays” and clarified the first sentence of the method as follows (lines 129-132, revised manuscript): Then, to evaluate DNA-damage and oxidative stress, the cell

  1. Figures 6 and 7 – have you got data to compare control and PL with FBS – as shown in other figures to demonstrate consistency in your research?

 Response3: Thank you for this insightful suggestion. In the murine wound-repair experiment, we compared platelet lysate (PL)-treated collagen/gelatin sponges with saline-treated controls only. We did not perform fetal bovine serum (FBS) group for two main reasons:

  1. Xenogeneic setting: For mice, both human PL and bovine FBS are xenogeneic. Directly comparing two xenogeneic dressings would not improve physiological relevance; instead, we chose saline as the neutral reference.
  2. Ethical considerations (3R principle): To minimize animal numbers and avoid unnecessary procedures, we limited the study to the two most informative groups.

We agree that an additional FBS comparator could enhance consistency with our in vitro data. We will therefore consider incorporating an FBS group in future in vivo studies designed specifically to address this point.

  1. Please provide a schematic of the potential mechanism you think PL uses to have the advantages it does over FBS.

 Response 4: We have added a graphical abstract that concisely illustrates the enhanced efficacy of platelet lysate (PL) on hASCs.

  1. Please add a paragraph of limitations to your conclusion – lack of female mice investigation, which implies any findings may be applicable only to cells from male donors; if this approach was to be taken forward.

Response 5: We have inserted a new Limitations paragraph in the Discussion section (lines 535–541 of the revised manuscript): “Several limitations should be noted. First, all platelet-lysate preparations were restricted to the pooled, GMP-grade product PLTMax®, and the efficacy of autologous or laboratory-prepared PL remains to be confirmed. Second, PLTMax® was applied xenogeneically in the murine model; validation using mouse-derived PL—or other syngeneic systems—will be required to strengthen translatability. Third, the in-vivo experiments were performed exclusively in male mice, and potential sex-dependent differences should be assessed in future studies that include female animals.”

Reviewer 2 Report

Comments and Suggestions for Authors

Authors’ Comments

  1. Methods: While the use of PLTMax suggests reliance on commercially available platelet lysate preparation, the manuscript lacks a detailed description of the isolation and characterization protocol for the PL employed in the culture experiments. Such methodological transparency is imperative for reproducibility and interpretability of the findings.

  1. To render the investigation more comprehensive and to robustly substantiate the enrichment of terms related to cellular protection and proliferative capacity, it would be prudent for the authors to perform additional enrichment analyses of the differentially expressed genes (DEGs) distinguishing the PL and FBS conditions. In particular, the incorporation of hierarchical clustering dendrograms, principal component analysis (PCA), and supplementary pathway analyses such as Gene Set Enrichment Analysis (GSEA) or Over-Representation Analysis (ORA) would greatly enhance the depth of interpretation.

  1. Figures 3 and 5: The fluorescent micrographs presented require substantial improvement in terms of resolution, magnification, and overall image quality. Moreover, appropriate and clearly defined scale bars should be included in all sub-panels to ensure proper spatial context and quantitative rigor.

  1. With respect to the wound healing assay, an investigation into the molecular mediators upregulated during the regenerative process would be highly informative. In this regard, the inclusion of quantitative PCR analyses assessing mRNA expression levels of genes associated with inflammation, cellular proliferation, and angiogenesis could yield valuable mechanistic insights and further substantiate the proposed therapeutic advantage conferred by PL.

Author Response

1. Methods: While the use of PLTMax suggests reliance on commercially available platelet lysate preparation, the manuscript lacks a detailed description of the isolation and characterization protocol for the PL employed in the culture experiments. Such methodological transparency is imperative for reproducibility and interpretability of the findings.

Response1: We appreciate the reviewer’s emphasis on methodological transparency. In our study all platelet-lysate experiments were conducted exclusively with the commercial product PLTMax® (Merck Millipore); no in-house preparation or additional processing was performed. We have added a clarifying sentence at the beginning of the Materials and Methods section (lines 99-100 of the revised manuscript) and in the list of abbreviations:
In all experimental sections of this study, PL refers to the commercial pooled human platelet lysate PLTMax® (Merck Millipore, Darmstadt, Germany).

Furthermore, we have inserted a new Limitations paragraph in the Discussion section (lines 535–537 of the revised manuscript): “First, all platelet-lysate preparations were restricted to the pooled, GMP-grade product PLTMax®, and the efficacy of autologous or laboratory-prepared PL remains to be confirmed.”

2. To render the investigation more comprehensive and to robustly substantiate the enrichment of terms related to cellular protection and proliferative capacity, it would be prudent for the authors to perform additional enrichment analyses of the differentially expressed genes (DEGs) distinguishing the PL and FBS conditions. In particular, the incorporation of hierarchical clustering dendrograms, principal component analysis (PCA), and supplementary pathway analyses such as Gene Set Enrichment Analysis (GSEA) or Over-Representation Analysis (ORA) would greatly enhance the depth of interpretation.

Response2: We appreciate this thoughtful recommendation and agree that complementary bio-informatic approaches—such as hierarchical clustering dendrograms, principal-component analysis (PCA), and pathway-level enrichment tests (e.g., GSEA or ORA)—would further refine interpretation of the RNA-seq dataset. In the current study our primary objective was to establish proof-of-concept evidence that platelet lysate (PL) promotes cytoprotective and proliferative programmes relative to fetal bovine serum (FBS). Accordingly, we focused on differential gene expression analysis followed by Gene Ontology over-representation tests (Results, Fig. 1), which already highlighted pathways associated with cellular protection and growth. Because of space limitations and the exploratory nature of this work, we have not included the additional multivariate and pathway analyses suggested. However, the raw sequencing data have been deposited and we are planning a follow-up bio-informatic study that will incorporate hierarchical clustering, PCA, and GSEA to build on the reviewer’s advice. We believe the current results sufficiently support our main conclusions, but we acknowledge that the proposed analyses will provide greater mechanistic depth and will be implemented in future investigations. We trust this explanation addresses the reviewer’s concern and clarifies the scope of the present manuscript.

3.Figures 3 and 5: The fluorescent micrographs presented require substantial improvement in terms of resolution, magnification, and overall image quality. Moreover, appropriate and clearly defined scale bars should be included in all sub-panels to ensure proper spatial context and quantitative rigor.

Response3: We have implemented the requested improvements:

  • Figure 5a replaced: To enhance clarity, the entire Figure 5a panel has been substituted with a new, higher-magnification, higher-resolution micrograph set.
  • Scale bars added: Clear scale bars are now present in every image of Figure 3a and Figure 5a.
  • Legend updated: The exact length of each scale bar is specified in the legends of Figures 3a and 5a.

4. With respect to the wound healing assay, an investigation into the molecular mediators upregulated during the regenerative process would be highly informative. In this regard, the inclusion of quantitative PCR analyses assessing mRNA expression levels of genes associated with inflammation, cellular proliferation, and angiogenesis could yield valuable mechanistic insights and further substantiate the proposed therapeutic advantage conferred by PL.

Response4: Thank you for this helpful suggestion. Our current manuscript has already included transcriptional evidence for proliferation and inflammation, and we briefly explained angiogenesis below.

  • Cell-proliferation markers – Lines 238–239 (Section 3.1) state:
    “We identified representative genes involved in cell proliferation (CDC20, CEP55, FEN1, MT2A and TPX2) (Fig. 1d).”
  • Inflammatory markers – Lines 337–339 (Section 3.4) report that IL-1β and IL-6 mRNA levels were lower in PL-treated hASCs than in the FBS group (Fig. S4).
  • Angiogenesis markers – Re-inspection of our RNA-seq dataset showed a modest upregulation of VEGF transcripts in the PL group. Because the primary focus of the present paper was cytoprotection and proliferation, and space for additional figures was limited, we did not include these angiogenic data in the revised text. The raw sequencing files containing this information are publicly available under BioProject accession PRJDB20462 (DDBJ) and can be explored in future, dedicated analyses.

Reviewer 3 Report

Comments and Suggestions for Authors

A good paper that should be published asap. Content is finally discriminating beweeen cell cultures in FBS and PL. PL is GMP compliant and thus needs to be introduces in all GMP related cell expansions. Let's finally forget FBS and start patient-oriented cell cultures. This paper demonstrates all these aspects.

Author Response

  1. A good paper that should be published asap. Content is finally discriminating beweeen cell cultures in FBS and PL. PL is GMP compliant and thus needs to be introduces in all GMP related cell expansions. Let's finally forget FBS and start patient-oriented cell cultures. This paper demonstrates all these aspects.

Response1: We gratefully acknowledge the reviewer’s encouraging evaluation and are delighted that our work is considered a valuable contribution toward patient-oriented, GMP-compliant cell culture. Thank you for your positive feedback.

Round 2

Reviewer 2 Report

Comments and Suggestions for Authors

All the priority comments were made in the first review.

Author Response

Editor’s remark / Reviewer 2 comment
“Reviewer #2 requested additional bioinformatic analyses, which have not been performed by the authors. I agree with the reviewer that the existing data should be analyzed in more depth to provide additional mechanistic insights. This should not be too much of an effort since the data already exist.”

Response:
We appreciate this recommendation and have now incorporated the requested bioinformatic analyses:

  1. Multivariate overview – Principal‑component analysis and hierarchical clustering have been added to the manuscript (Fig. 1a,b). The corresponding methods are detailed in lines 112–116 and the results are described in lines 220–226. These analyses confirm tight clustering within each treatment group and clear separation among Control, FBS, and PL samples.
  2. Pathway‑level ORA expansion – In addition to the original GO‑BP ORA, we have performed KEGG and Reactome ORA on the same DEG lists. These results are summarized in the Results section (lines 251–253) and presented in Supplementary Fig. S3. The enriched pathways—chiefly cell‑cycle—support the mechanistic conclusions drawn from the GO analysis.

Round 3

Reviewer 2 Report

Comments and Suggestions for Authors

The authors have addressed the majority of my previous recommendations, and I have no further comments.